# Hyperglycemia and Hypoglycemia Are Associated with In-Hospital Mortality among Patients with Coronavirus Disease 2019 Supported with Extracorporeal Membrane Oxygenation

**DOI:** 10.3390/jcm11175106

**Published:** 2022-08-30

**Authors:** Kuk Hui Son, Woong-Han Kim, Jae Gun Kwak, Chang-Hyu Choi, Seok In Lee, Ui Won Ko, Hyoung Soo Kim, Haeyoung Lee, Euy Suk Chung, Jae-Bum Kim, Woo Sung Jang, Jae Seung Jung, Jieon Kim, Young Kyung Yoon, Seunghwan Song, Minji Sung, Myung Hun Jang, Young Sam Kim, In-Seok Jeong, Do Wan Kim, Tae Yun Kim, Soon Jin Kim, Su Wan Kim, Joonhwa Hong, Hyungmi An

**Affiliations:** 1Department of Thoracic and Cardiovascular Surgery, Gachon University Gil Medical Center, Gachon University College of Medicine, Incheon 21565, Korea; 2Department of Thoracic and Cardiovascular Surgery, Seoul National University Children’s Hospital, Seoul National University College of Medicine, Seoul 03080, Korea; 3Pulmonary and Allergy Division, Department of Internal Medicine, Gachon University Gil Medical Center, Gachon University College of Medicine, Incheon 21565, Korea; 4Department of Thoracic and Cardiovascular Surgery, Hallym Sacred Heart Hospital, Hallym University College of Medicine, Chuncheon 24252, Korea; 5Department of Thoracic and Cardiovascular Surgery, Kosin University Gospel Hospital, Kosin University College of Medicine, Busan 49267, Korea; 6Department of Thoracic and Cardiovascular Surgery, Kangbuk Samsung Hospital, Sungkyunkwan University, Seoul 03181, Korea; 7Department of Thoracic and Cardiovascular Surgery, Keimyung University Dongsan Hospital, Keimyung University School of Medicine, Daegu 42601, Korea; 8Department of Thoracic and Cardiovascular Surgery, Korea University College of Medicine, Seoul 02841, Korea; 9Division of Infectious Disease, Department of Internal Medicine, Korea University College of Medicine, Seoul 02841, Korea; 10Department of Thoracic and Cardiovascular Surgery, Pusan National University Hospital, Biomedical Research Institute, Pusan National University School of Medicine, Busan 49241, Korea; 11Health Convergence Medicine Laboratory, Biomedical Research Institute, Pusan National University Hospital, Busan 49241, Korea; 12Department of Rehabilitation Medicine, Biomedical Research Institute, Pusan National University Hospital, Busan 49241, Korea; 13Department of Thoracic and Cardiovascular Surgery, Inha University Hospital, Inha University School of Medicine, Incheon 22332, Korea; 14Department of Thoracic and Cardiovascular Surgery, Chonnam National University Hospital, Chonnam National University Medical School, Gwangju 61649, Korea; 15Department of Thoracic and Cardiovascular Surgery, Jeonbuk National University Hospital, Jeonju 54907, Korea; 16Department of Thoracic and Cardiovascular Surgery, Jeju National University Hospital, Jeju National University School of Medicine, Jeju 63241, Korea; 17Department of Thoracic and Cardiovascular Surgery, Chung-Ang University Hospital, Seoul 06973, Korea; 18Institute of Convergence Medicine, Ewha Womans University Mokdong Hospital, Seoul 07985, Korea

**Keywords:** COVID-19, diabetes, hyperglycemia, hypoglycemia, extracorporeal membrane

## Abstract

Metabolic abnormalities, such as preexisting diabetes or hyperglycemia or hypoglycemia during hospitalization aggravated the severity of COVID-19. We evaluated whether diabetes history, hyperglycemia before and during extracorporeal membrane oxygenation (ECMO) support, and hypoglycemia were risk factors for mortality in patients with COVID-19. This study included data on 195 patients with COVID-19, who were aged ≥19 years and were treated with ECMO. The proportion of patients with diabetes history among nonsurvivors was higher than that among survivors. Univariate Cox regression analysis showed that in-hospital mortality after ECMO support was associated with diabetes history, renal replacement therapy (RRT), and body mass index (BMI) < 18.5 kg/m^2^. Glucose at admission >200 mg/dL and glucose levels before ventilator >200 mg/dL were not associated with in-hospital mortality. However, glucose levels before ECMO >200 mg/dL and minimal glucose levels during hospitalization <70 mg/dL were associated with in-hospital mortality. Multivariable Cox regression analysis showed that glucose >200 mg/dL before ECMO and minimal glucose <70 mg/dL during hospitalization remained risk factors for in-hospital mortality after adjustment for age, BMI, and RRT. In conclusion, glucose >200 mg/dL before ECMO and minimal glucose level <70 mg/dL during hospitalization were risk factors for in-hospital mortality among COVID-19 patients who underwent ECMO.

## 1. Introduction

The negative impact of diabetes on morbidity and mortality has been reported in coronavirus disease 2019 (COVID-19). Recently reported data on patients with COVID-19 showed that two-thirds of the mortality was exhibited by patients with diabetes [1]. Several studies have shown that diabetes also affects the severity of COVID-19 [2,3,4,5]. Patients with COVID-19 and diabetes showed a greater rate of organ damage, a tendency toward hypercoagulation, and higher levels of inflammation-related factors [2]. A study from China including 605 patients with COVID-19 showed that patients with a fasting blood glucose (FBG) level of 110–124 mg/dL, which was measured at admission, had an odds ratio for morbidity of 28 days, approximately more than three times that of those with FBG <110 mg/dL [6]. Patients with secondary hyperglycemia, defined as blood glucose levels >180 mg/dL and HbA1c levels <6.5% without a history of diabetes, showed increased mortality as compared to patients with normal glucose levels [7].

Hypoglycemia is also related to increased mortality in patients admitted to the intensive care unit (ICU) [8]. A multicenter, retrospective study that included 1544 COVID-19 patients showed that hypoglycemia, defined as blood glucose <70 mg/dL, was associated with mortality [9].

Extracorporeal membrane oxygenation (ECMO) is widely used to treat severe acute respiratory distress syndrome (ARDS) in COVID-19 [10,11,12,13,14]. Although the risk factors for mortality after ECMO support in patients with COVID-19 have not been clearly elucidated, older age [14,15,16,17,18,19,20], partial pressure of arterial oxygen over the fractional inspired oxygen (PaO_2_/FiO_2_) ratio [11,14,15,21], and renal replacement therapy (RRT) [11,14,21] have been associated with increased mortality.

Although preexisting diabetes, hyperglycemia, and hypoglycemia are known to increase disease severity and mortality in COVID-19, it has not been fully explored whether dysglycemia, such as hyperglycemia or hypoglycemia, are associated with mortality after ECMO in COVID-19 patients.

We evaluated whether diabetes history, hyperglycemia before and during ECMO support, and hypoglycemia were risk factors for mortality in COVID-19 patients supported with ECMO in Korea.

## 2. Materials and Methods

### 2.1. Data Source

A nationwide registry for Korean COVID-19 ECMO patients was established in 2021 and patient registration is still in progress. This registry is operated by The Korean Society for Thoracic and Cardiovascular Surgery and is supported by the Korea Disease Control and Prevention Agency. Inclusion criteria for the registry are COVID-19 patients aged ≥19 years who were treated with ECMO. COVID-19 was diagnosed with severe acute respiratory syndrome coronavirus 2 (SARS-CoV-2) using real-time polymerase chain reaction (RT-PCR) of sputum or nasopharyngeal swab samples. Up to the current date, 13 hospitals have participated in the registry and those hospitals are located in almost all parts of Korea.

Eligibility for ECMO, patient management during ECMO, and weaning from ECMO were followed by Extracorporeal Life Support Organization (ELSO) guidelines [22].

This study was approved by the Institutional Review Board (IRB) of each participating hospital. Informed consent was obtained according to IRB guidelines. The requirement for informed consent was waived when patients were discharged before IRB approval in each hospital. Therefore, data from the patients who waived their consent were retrospectively collected. Clinical data for the registry were collected retrospectively and prospectively, and all hospitals entered the data into a predesigned common-case record form. The data collected included patient demographics, medical history, smoking, comorbid conditions, COVID-19 related symptoms, laboratory tests (at admission, before the start of mechanical ventilation, and before ECMO initiation), maximum and minimum levels of laboratory parameters during hospitalization, sequential organ failure assessment (SOFA), mechanical ventilation factors, adjunctive therapies, such as prone positioning during ECMO support, and ECMO outcomes.

### 2.2. Data Collection

This study is a multicenter retrospective study with data derived from the Korean COVID-19 ECMO registry. For this study, data from 195 patients admitted between February 2020 and May 2022 were used.

### 2.3. Statistical Analysis

Continuous variables were presented using the mean (standard deviation, SD) or median (interquartile range, IQR). Counts (N) and percentages (%) were used for categorical variables.

Mean differences between survivor group and nonsurvivor group were assessed by Student’s *t*-test for normally distributed continuous variables and Mann–Whitney U test for skewed continuous variables. For categorical variables, Pearson χ^2^ test was performed to compare proportions between two groups.

The follow-up period was defined as the date of ECMO initiation till the patient was treated as a censoring event. Patients who were discharged alive were censored on the date of hospital discharge; patients transferred to another hospital were censored on the date of transfer; and patients whose outcomes were not finalized were censored on the date of final data collection for this study (15 June 2022). The minimum and maximum follow-up times were 0 and 499 days after ECMO initiation, respectively.

We applied the Cox proportional hazard model to assess the risk of in-hospital mortality for COVID-19 patients while taking ECMO support. Possible risk factors were selected based on differences in baseline characteristics or laboratory data. All variables tested in the Cox regression analysis were categorized. The SOFA score and initial diastolic blood pressure (BP) were dichotomized by their cut-off value for mortality. The cut-off was chosen based on the receiver operator characteristic curve and Youden index. History of diabetes was defined as when patients had a documented diagnosis for diabetes before admission or a prescription for diabetes medication. According to the American Diabetes Association, hyperglycemia is defined in hospitalized patients as a blood glucose level >140 mg/dL [23,24]. Diabetes is defined as a random plasma glucose level of ≥200 mg/dL with classic symptoms of hyperglycemia [25]. Therefore, we categorized various glucose parameters according to the definition of hyperglycemia (>140 mg/dL or >200 mg/dL) and evaluated the association between hyperglycemia and in-hospital mortality after ECMO support in patients with COVID-19. Body mass index (BMI) was classified as three categories (<18.5, 18.5–24.9, ≥25 kg/m^2^) according to the World Health Organization Asian classification [26].

A hazard ratio (HR) for each variable was calculated with associated 95% confidence interval (CI) and a statistical significance threshold of *p* < 0.05. Proportional hazard assumption was visually assessed using log(−log) plots. In the multivariable analysis, adjusted confounders for calculating the HR of glucose parameters for mortality were age (19–49, 50–59, 60–69, ≥70 years old), RRT before ECMO start, and BMI. To avoid the multicollinearity issue between preexisting diabetes and glucose parameters, we made four multivariable Cox models for calculating HR of each glucose parameter (glucose before ventilator >200 mg/dL, glucose before ECMO >200 mg/dL, minimal glucose <70 mg/dL, and diabetes history).

For further analysis when evaluating the effect of a history of diabetes and glucose level before ECMO on in-hospital mortality, participants in our study were categorized into four groups according to the history of diabetes and glucose level ≤ 200 mg/dL or >200 mg/dL before ECMO. In addition, the effect of diabetes history and minimal glucose during hospitalization on in-hospital mortality was evaluated by categorizing patients into four groups according to the history of diabetes and minimal glucose ≥ 70 mg/dL or <70 mg/dL.

There were incomplete data, especially data from patients who were referred. The amount of missing data is reported in Appendix A. Missing data were handled by list-wise deletion.

All statistical analyses were performed using SPSS version 22 (IBM Corp., Armonk, NY, USA). Cumulative incidence of in-hospital mortality from the time of ECMO initiation was estimated by R Version 4.0.4 (The R Foundation for Statistical Computing, Institute for Statistics and Mathematics, Vienna, Austria).

## 3. Results

### 3.1. Characteristics and Outcome of the Patients

Among 195 COVID-19 ECMO patients, 110 (54.6%) died in hospital. We estimated the distribution of ECMO duration and time to in-hospital death using Kaplan–Meier estimators. The median follow-up period was 40 (20–64) days.

The estimated cumulative incidence of in-hospital mortality 240 days after the initiation of ECMO was 82.9% (95% CI 68.7–90.7) (Figure 1). Cause of death was multiorgan failure (*n* = 22), neurological complication (*n* = 3), septic shock (*n* = 58), and others such as bleeding or lung fibrosis (*n* = 27).

Patient characteristics and laboratory data were compared between the survivor and nonsurvivor groups (Appendix A). Among those, variables which showed a difference between the survivor and nonsurvivor groups were analyzed by Cox regression models.

### 3.2. Evaluation of Risk Factors for In-Hospital Mortality after ECMO Support

To evaluate which independent variables were associated with in-hospital mortality after ECMO support, Cox regression analysis was performed (Table 1). Univariate analysis indicated that in-hospital mortality was increased by age. History of diabetes was associated with in-hospital mortality. Smoking history and an initial diastolic BP of <80 mmHg were not associated with in-hospital mortality.

Obesity (BMI ≥ 25 kg/m^2^) was not associated with in-hospital mortality; however, underweight (BMI < 18.5 kg/m^2^) was associated with in-hospital mortality compared to normal BMI (18.5–24.9 kg/m^2^).

In-hospital mortality was not associated with ventilator use for ECMO for more than 7 days. RRT before ECMO was associated with in-hospital mortality. A SOFA score >8 was not associated with mortality.

An initial glucose level >200 mg/dL was not associated with in-hospital mortality. However, glucose levels >200 mg/dL before ventilator support and glucose levels >200 mg/dL before ECMO were associated with in-hospital mortality. A maximum glucose level >200 mg/dL was not associated with in-hospital mortality. Initial glucose >140 mg/dL, maximum glucose >140 mg/dL, glucose before ventilator support >140 mg/dL, and glucose before ECMO >140 mg/dL were not associated with mortality. A minimal glucose level of <70 mg/dL was associated with mortality.

Appendix A and Appendix A showed results of multivariable Cox regression analysis. After adjustment for age (<50, 50–59, 60–69, and ≥70 years), BMI, and RRT before ECMO, glucose >200 mg/dL before ventilator use (model 1) and diabetes history (model 4) were not associated with in-hospital mortality. However, glucose >200 mg/dL before ECMO (model 2) and minimal glucose level <70 mg/dL (model 3) were associated with in-hospital mortality after adjustment for age, BMI, and RRT before ECMO.

Table 2 and Appendix A showed the effect of diabetes history and glucose level before ECMO on in-hospital mortality. No diabetes history and glucose ≤200 mg/dL before ECMO, and with diabetes history and glucose ≤200 mg/dL before ECMO were not associated with in-hospital mortality. However, the HR of no diabetes history and glucose >200 mg/dL before ECMO was 1.97 (95% CI, 1.00–3.83; *p* < 0.05). The HR of diabetes history and glucose before ECMO >200 mg/dL was even higher (HR, 2.16; 95% CI, 1.17–3.97; *p* = 0.01).

Table 3 and Appendix A showed the effect of diabetes history and minimal glucose during hospitalization on in-hospital mortality. When the reference was no diabetes history and minimal glucose level ≥70 mg/dL, diabetes history and minimal glucose level <70 mg/dL were not associated with in-hospital mortality. However, the HR of no diabetes history and minimal glucose ≥70 mg/dL was 6.66 (95% CI, 3.48–12.75; *p* < 0.05). The HR of diabetes history and minimal glucose <70 mg/dL was 2.82 (95% CI, 1.24–6.42; *p* = 0.01).

## 4. Discussion

The main finding of our study was that hyperglycemia (glucose levels >200 mg/dL before ECMO) and hypoglycemia (minimal glucose level <70 mg/dL during hospitalization) were associated with in-hospital mortality of COVID-19 patients who underwent ECMO.

Diabetes leads to various impairments in the innate, adaptive, and acquired immune systems [27,28]. Both the activated and resting states of neutrophils from patients with diabetes exhibit an extremely increased secretion of tumor necrosis factor (TNF)-α, interleukin (IL)-8, and IL-1β, and elevated levels of inflammatory cytokines are associated with increased susceptibility to pathogens [29].

SARS-CoV-2 has been reported to cause apoptosis of lymphocytes, such as CD3, CD4, and CD8+ T cells, thus decreasing circulating immune cells and causing lymphocytopenia [30]. Therefore, these changes in the immune system caused by SARS-CoV-2 are aggravated in diabetes, which eventually decreases the host defense against SARS-CoV-2 and increases susceptibility to SARS-CoV-2 infection [31]. SARS-CoV-2 infection leads to increased levels of inflammatory cytokines such as TNF-α, IL-1, and IL-6, which are also increased in diabetes [32,33]. These cytokines cause a cytokine storm, resulting in multiple organ damage; therefore, patients with COVID-19 and diabetes showed a more severe prognosis for COVID-19 than patients without diabetes [34]. Diabetes or the hyperglycemic state is related to increased oxidative stress, which is induced by increased generation of reactive oxygen species (ROS) or the decreased antioxidant defense mechanism [35]. Increased ROS leads to the failure of the antioxidant mechanism that is essential to abolish viral replication [36]. In fact, increased oxidative stress is associated with mortality in patients with COVID-19 [36].

A meta-analysis reported that 8 ± 6% of 46,248 patients with COVID-19 had diabetes (95% CI, 6–11%) [37]. Pooled data from 10 studies that included 2,209 patients with COVID-19 showed that 11% of patients had diabetes [38]. A retrospective study from China that included 201 patients with COVID-19 reported that the HR of diabetes for the development of ARDS was 2.34 (95% CI, 1.35–4.05; *p* = 0.02) [39].

Similar to previous studies, our univariate Cox regression analysis revealed that a history of diabetes was associated with in-hospital mortality. However, a history of diabetes was not associated with in-hospital mortality after adjusting for age, BMI, and RRT before ECMO. A previous study showed that old age (≥40 years) was a risk factor for hospitalization in type 1 diabetes patients with COVID-19, since older patients had greater comorbidities [40]. This might be an explanation as to why the association between a history of diabetes and in-hospital mortality disappeared after adjustment for age in our study.

In addition to diabetes, hyperglycemia has been associated with increased mortality in COVID-19. A retrospective study from China that included 461 patients with COVID-19 showed that hyperglycemia (any glucose level during hospitalization >140 mg/dL) was positively associated with inflammation levels, such as procalcitonin, C-reactive protein, erythrocyte sedimentation rate, and COVID-19 severity [41].

After adjustment for age, BMI, and RRT before ECMO, glucose levels >200 mg/dL before ventilator use were not significant risk factors for in-hospital mortality in our study. However, glucose levels >200 mg/dL before ECMO were still associated with in-hospital mortality even after adjustment for age, BMI, and RRT before ECMO. When HR was evaluated according to the categorized groups for history of diabetes and glucose levels before ECMO, the HR of diabetes history and glucose levels >200 mg/dL before ECMO was higher than those with no history of diabetes and glucose levels >200 mg/dL before ECMO. This suggests that patients with a history of diabetes were more highly associated with mortality than patients who had no history of diabetes when their blood glucose level before ECMO was >200 mg/dL.

To maintain normal glucose levels, cardiovascular function and the central nervous system should act normally [42,43]. Under stressful conditions, such as critical illness, excess secretion of counter-regulatory insulin hormones, including glucagon and cortisol, leads to changes in carbohydrate metabolism by increasing insulin resistance, enhancing hepatic glucose synthesis, and decreasing peripheral glucose utilization [43,44]. Increased epinephrine levels due to stress inhibit insulin secretion from β-cells and promote glucagon secretion [44,45]. Moreover, pro-inflammatory cytokines such as TNF-α and IL-6 interfere with the insulin signaling pathway and increase insulin resistance under stress conditions [46,47]. Therefore, stress hyperglycemia can be induced by critical illness and aggravates the disease. It is debatable whether stress hyperglycemia is a causally related factor for poor outcomes, such as mortality, or whether it is just a marker of disease severity [48].

A complete explanation of why a glucose level >200 mg/dL before ECMO was the most significant risk factor for in-hospital mortality among other glucose parameters was not possible, as our study is a registry study. It appeared that glucose level > 200 mg/dL before ECMO most sensitively reflected the stress hyperglycemia condition in patients with COVID-19 supported by ECMO, among other glucose parameters. It is also known that all types of artificial extracorporeal circulation devices, including ECMO, induce increased pro-inflammatory signaling pathways by initiating blood interactions with artificial surfaces [49,50]. Moreover, an animal study showed that ECMO circulation leads to increased levels of pro-inflammatory cytokines, such as TNF-α and IL-6, especially in diabetic animals, rather than in animals with normal glucose levels [51]. Thus, it is possible that patients with higher glucose levels before ECMO were more affected by ECMO, which induced a greater increase in inflammatory cytokines than in patients with normal glucose levels before ECMO. In other words, the glucose level >200 mg/dL before ECMO suggests that patients may be in a more vulnerable state in which ECMO-induced inflammation impacts more severely than in patients who have blood glucose levels lower than 200 mg/dL. In another aspect, hyperglycemia was associated with in-hospital mortality; however, it might not be a strong prognostic factor, since other hyperglycemia variables such as glucose levels >200 mg/dL before ventilator were not associated with in-hospital mortality after adjustment for age, BMI, and RRT before ECMO. It is known that right ventricular (RV) systolic and diastolic dysfunction is associated with poor glycemic control in type 2 diabetes [52]. RV dysfunction, which was evaluated with the RV longitudinal strain (RVLS), was related to mortality in patients with COVID-19 [53]. Thus, it is also possible that hyperglycemia could be an indirect risk factor for in-hospital mortality that is related to RV dysfunction. Since we did not collect echo data in this study, we must evaluate whether hyperglycemia directly affects in-hospital mortality or whether it is indirectly affected through RV dysfunction in the future study. Moreover, recent meta-analysis showed that incidence of diabetes in post-COVID-19 patients was higher than healthy control subjects [54]. Among patients who recovered from COVID-19, 42% showed subclinical RV dysfunction, which was evaluated with RVLS [55]. Thus, it might be possible that hyperglycemia is associated with RV dysfunction even in the recovery period of COVID-19 or post COVID-19 syndrome. Those possible relationships should be evaluated in the future study.

Although hypoglycemia in diabetic patients is frequently induced by insulin or hypoglycemic agents, hypoglycemia can occur in non-diabetic patients when they are hemodynamically unstable [56,57,58]. Decreased renal function, hypothyroidism, and disturbed glucose regulation mechanisms such as reduced glucagon and increased adrenalin release, inadequate glucose supply, and strict glycemic control management are risk factors for hypoglycemia [59,60,61]. Hypoglycemia aggravates the increase in pro-inflammatory cytokines, which could induce a cytokine storm [62]. In COVID-19 patients, hypoglycemia is related to increased mortality [63].

Patients who underwent veno-arterial ECMO for the treatment of cardiogenic shock or cardiac arrest showed the best survival when their blood glucose level at cannulation for ECMO start was 140–240 mg/dL as compared to hypoglycemia (<80 mg/dL), (80–140 mg/dL), moderate hyperglycemia (241–400 mg/dL), and severe hyperglycemia (>400 mg/dL) [64].

The most frequently used glucose level for defining hypoglycemia is a plasma glucose level less than 70 mg/dL [65]. Thus, we defined hypoglycemia as glucose level <70 mg/dL. In our study, univariate Cox regression analysis showed that hypoglycemia was associated with in-hospital mortality. Moreover, hypoglycemia was still a predictor of in-hospital mortality, even after adjustment for age, BMI, and RRT before ECMO. When HR was evaluated according to categorized groups for history of diabetes and hypoglycemia, HR of no diabetes history and minimal glucose <70 mg/dL was higher than the HR of diabetes history and minimal glucose <70 mg/dL. This suggests that patients without a history of diabetes were more highly associated with mortality than patients with a history of diabetes, when they were exposed to hypoglycemia below 70 mg/dL.

A representative randomized control study showed that strict glucose control targeting glucose levels below 110 mg/dL increased hypoglycemic events as compared to conventional blood glucose control management, in which blood glucose was targeted below 180 or 200 mg/dL [59]. Based on this study, it is suggested that the blood glucose level of ICU-admitted patients should be controlled at approximately 180 or 200 mg/dL. However, there are no suggested blood glucose levels for COVID-19 patients who underwent ECMO. Our study showed that both hyperglycemia and hypoglycemia were associated with in-hospital mortality in COVID-19 patients who underwent ECMO. Moreover, hypoglycemia was even more harmful in patients with no history of diabetes. The targeted glucose control level for COVID-19 patients who underwent ECMO should be evaluated in future studies to decrease mortality.

This study has several limitations. First, this study had a small sample size; therefore, it was impossible to deduce a cause-effect relationship between hyperglycemia or hypoglycemia and mortality. Thus, it is essential to validate whether hyperglycemia or hypoglycemia are prognostic factors for in-hospital mortality of COVID-19 patients who underwent ECMO by using another larger size cohort in a future study. Second, there were significant missing data, since many patients were referred for ECMO under mechanical ventilation. In particular, there were significant missing data related to glucose parameters, which is a main limitation of our study. Missing data could cause bias; thus, our present results should be validated with other cohorts. However, our results could show importance as to why clinicians should properly manage glucose parameters during ECMO as real-world data. Third, history of diabetes was defined only by a medical history of diabetes and diabetes medication without the results of HbA1c, since HbA1c was not available for all patients at admission. Fourth, the glucose parameters in our study could be fasting or random depending on the patient. These factors might act as biases; however, they reflect the real-world conditions in which most blood samples in patients with COVID-19 were obtained, regardless of fasting. For decreasing those biases, history of diabetes should be defined based on HbA1c, and blood samples for measuring glucose parameters should be performed under the fasting state in the future study.

## 5. Conclusions

In conclusion, glucose levels >200 mg/dL before ECMO and minimal glucose level <70 mg/dL during hospitalization were better predictors of in-hospital mortality than other glucose parameters, such as initial glucose >140 mg/dL or 200 mg/dL and glucose before ventilator use >140 mg/dL or 200 mg/dL.

## Figures and Tables

**Figure 1 jcm-11-05106-f001:**
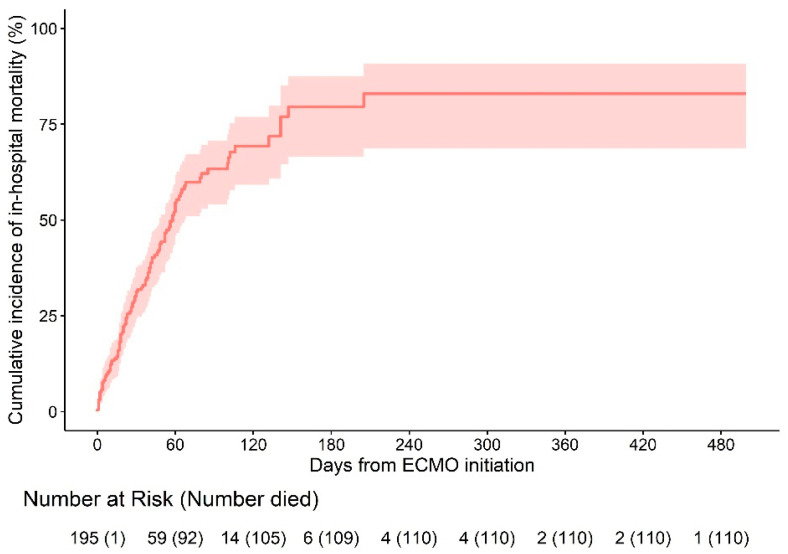
Cumulative incidence of in-hospital mortality from the time of ECMO initiation.

**Table 1 jcm-11-05106-t001:** Univariate Cox regression analysis for mortality after ECMO support in COVID-19.

Variable	HR	95% CI	*p*-Value
Age, years			
19–49	Reference		
50–59	1.42	0.73–2.56	0.30
60–69	1.84	1.05–3.23	0.03
≥70	1.91	0.98–3.72	0.05
History of diabetes	1.50	1.01–2.21	0.04
Smoking			
Current smoker	2.57	0.83–8.00	0.10
Ex-smoker	2.19	0.80–6.02	0.13
Never smoker	Reference		
Initial diastolic BP <80 mmHg	1.30	0.88–1.91	0.18
BMI, kg/m^2^			
<18.5	7.87	1.83–33.79	<0.01
18.5–24.9	Reference		
≥25	0.75	0.51–1.10	0.13
Ventilator to ECMO >7 days	1.03	0.67–1.57	0.90
RRT before ECMO	2.47	1.42–4.28	<0.01
SOFA >8	1.54	0.95–2.49	0.06
Initial glucose >200 mg/dL	1.12	0.73–1.72	0.60
Glucose before ventilator >200 mg/dL	1.69	1.05–2.72	0.03
Glucose before ECMO >200 mg/dL	1.86	1.17–2.96	<0.01
Maximal glucose >200 mg/dL	0.70	0.31–1.62	0.41
Initial glucose >140 mg/dL	0.86	0.58–1.27	0.44
Glucose before ventilator >140 mg/dL	0.85	0.54–1.35	0.50
Glucose before ECMO >140 mg/dL	1.45	0.80–2.63	0.22
Minimal glucose <70 mg/dL	3.07	1.94–4.85	<0.01

BP, blood pressure; BMI, body mass index; ECMO, extracorporeal membrane oxygenation; SOFA, Sequential Organ Failure Assessment; RRT, renal replacement therapy; CI, confidence interval; HR, hazard ratio.

**Table 2 jcm-11-05106-t002:** Multivariable Cox regression analysis of DM history and glucose before ECMO.

Variable	HR	95% CI	*p*-Value
Age, years			
19–49	Reference		
50–59	1.04	0.39–2.76	0.94
60–69	1.90	0.84–4.29	0.13
≥70	2.53	1.00–6.39	0.049
BMI, kg/m^2^			
<18.5	9.82	2.01–47.99	<0.01
18.5–24.9	Reference		
≥25	0.79	0.47–1.31	0.36
RRT before ECMO	2.19	1.08–4.42	0.03
Combination of DM history and glucose before ECMO			
No DM and glucose before ECMO ≤200 mg/dL	Reference		
DM and glucose before ECMO ≤200 mg/dL	1.28	0.61–2.71	0.52
No DM and glucose before ECMO >200 mg/dL	1.97	1.00–3.83	0.047
DM and glucose before ECMO >200 mg/dL	2.16	1.17–3.97	0.01

DM, diabetes mellitus; ECMO, extracorporeal membrane oxygenation; RRT, renal replacement therapy; BMI, body mass index; HR, hazard ratio; CI, confidence interval.

**Table 3 jcm-11-05106-t003:** Multivariable Cox regression analysis of DM history and minimal glucose level during admission.

Variable	HR	95% CI	*p*-Value
Age, years			
19–49	Reference		
50–59	0.82	0.35–1.94	0.65
60–69	1.52	0.75–3.09	0.25
≥70	1.98	0.85–4.64	0.12
BMI, kg/m^2^			
<18.5	12.71	2.54–63.49	<0.01
18.5–24.9	Reference		
≥25	0.79	0.47–1.31	0.37
RRT before ECMO	3.17	1.57–6.41	<0.01
Combination of DM history and minimal glucose			
No DM and minimal glucose ≥70 mg/dL	Reference		
DM and minimal glucose ≥70 mg/dL	1.78	0.98–3.23	0.59
No DM and minimal glucose <70 mg/dL	6.66	3.48–12.75	<0.01
DM and glucose before ECMO <70 mg/dL	2.82	1.24–6.42	0.01

DM, diabetes mellitus; ECMO, extracorporeal membrane oxygenation; RRT, renal replacement therapy; BMI, body mass index; HR, hazard ratio; CI, confidence interval.

## Data Availability

Not applicable.

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
