# Peer review of "Hyperglycemia and Hypoglycemia Are Associated with In-Hospital Mortality among Patients with Coronavirus Disease 2019 Supported with Extracorporeal Membrane Oxygenation"

_jcm, 2022, doi:10.3390/jcm11175106_

Round 1

Reviewer 1 Report (Previous Reviewer 2)

This study aimed to evaluate whether preexisting diabetes, hyperglycemia before and during ECMO support, and hypoglycemia are risk factors for mortality in patients with COVID-19 supported with ECMO in Korea. The study could be useful for the management of severely ill patients with COVID-19. However, several aspects need to be clarified.

The following drawbacks, considered as major revisions, should be addressed to improve the paper:

Material and methods

-          The type of the study design should be given

-          A short description of the Korean COVID-19 ECMO registry is needed giving indication of the population covered. How are the 13 hospitals contributing to the registry geographically distributed?

-          Details on ECMO treatment are necessary: duration, discontinuation, related complications

Statistical analysis

-          Line 130: Mean and standard deviation are not appropriate indices to summarize variables with a non normal distribution. Please , modify appropriately. Moreover, don’t use ± after mean, it is not correct. Use brackets in the text and in the tables.

-          Line 132: what does authors mean for “controlled” or “uncontrolled seizure group”? please, clarify

-          The description of the follow up should be given before statistical analysis. Please, move lines between 137 and 142, before this paragraph.

-          Which was the minimum follow-up time to detect in-hospital deaths?

-          Line 144: Why did authors dichotomized all variables used in the Cox analysis? Cut-off values are used according to guide lines, if available, which represent standardized categories, i. e BMI classification of WHO. This will allow to compare results between studies.

-          Line 150-153: It is not a multivariate rather than a multivariable analysis. Please, modify.

-          Methods for goodness of fit of the model have to be reported and results in the Result section

Results

-          Line 159-162: there is no indication of the distribution of the follow up time, it should be added.

-          Table 1 and 2: the distribution of patients’ characteristics is evaluated according to the outcome, a time-dependent variable, therefore comparisons between groups should be performed through appropriate statistical methods, such as Kaplan-Meier analysis. In this case, number at risk and number of cases at each time can be added. Moreover, if variables are used classified in categories, these should be reported in the descriptive analysis.

-          Lines 266-267: Definition of preexisting diabetes is not clear, what authors mean by “diabetes medication use”? since when? How many prescriptions, etc.? Moreover, the definition should be moved to the Method section.

-          Table 3 will not be useful any more after an appropriate descriptive analysis.

-          Line 130: substitute multivariate with multivariable or multiple and throughout the text.

-          Lines 310-313: should be moved to the Statistical analysis section.

-          Lines 319-324: Move to the Statistical analysis section

-          Lines 332-334: Move to the Statistical analysis section

-          Table 4 and 5. Use only 1 cut off for small p-values, i. e <0.001. For those <0.05 give exact value.

-          Supplementary Table 1, Table 4 and Table 5. All adjusted multivariable Cox models have estimates with low precision as indicated by the 95% confidence intervals. Authors should give indication of the power of the models. Were interactions in these models explored, such as between age, BMI and diabetes and/or glucose levels?

Discussion

Overall, this section needs to be revised and reorganized. Authors should avoid to repeat concepts already expressed in the other section of the manuscript. Results of the study can be compared and discuss related to similar studies. In particular:

-          Lines 394-399: already used in the Introduction section, do not repeat.

-          Lines 400-407: is redundant with results, it is not useful to repeat it.

-          Lines 414-415: Results reported are not supported by those reported in the following lines 415-417.

-          Lines 422-423: the interpretation of the relative risk is not correct.

-          Lines 434-441: should be moved to methods section

-          Lines 442-447: Remove. Results of the adjusted analysis can be discussed.

-          Lines 489-490: should be removed as redundant with Introduction section

-          Lines 531-534: Authors should discuss how the mentioned limits could impact results for a better interpretation.

Author Response

Q1)The type of the study design should be given

--> We added study design as below;

This study is a multicenter retrospective study which data derived from Korean COVID-19 ECMO patients registry. For this study, data from 195 patients admitted between February 2020 and May 2022 was used.

Q2) A short description of the Korean COVID-19 ECMO registry is needed giving indication of the population covered. How are the 13 hospitals contributing to the registry geographically distributed?

--> We added information of Korean COVID-19 registry like below;

A nationwide registry for Korean COVID-19 ECMO patients was established in 2021 and patient registration is still in progress. This registry is operated by The Korean Society for Thoracic and Cardiovascular Surgery and is supported by the Korea Disease Control and Prevention Agency. Inclusion criteria of the registry is COVID-19 patients aged ≥ 19 years who were treated with ECMO. COVID-19 was diagnosed with severe acute respiratory syndrome coronavirus 2 (SARS-CoV-2) using real-time polymerase chain reaction (RT-PCR) of sputum or nasopharyngeal swab samples. Until now, 13 hospitals participated in the registry and those hospitals locate in almost all parts of Korea.

Q3) Details on ECMO treatment are necessary: duration, discontinuation, related complications

--> We added that information related with your recommendation as below;

Eligibility for ECMO, patient management during ECMO, and weaning from ECMO were followed by ELSO guidelines.

Statistical analysis

Q4)  Line 130: Mean and standard deviation are not appropriate indices to summarize variables with a non normal distribution. Please , modify appropriately. Moreover, don’t use ± after mean, it is not correct. Use brackets in the text and in the tables.

-->  As your recommendation we changed presentation form as blow;

Continuous variables were presented using the mean (standard deviation, SD) or median (interquartile ranger, IQR). Counts (N) and percentages (%) were used for categor-ical variables.

Q5) Line 132: what does authors mean for “controlled” or “uncontrolled seizure group”? please, clarify

--> It was wrong expression. We changed the sentence as below;

Mean differences between survivor group and nonsurvior group were assessed by Student’s t-test for normally distributed continuous variables and Mann Whitney U test for skewed continuous variables. For categorical variables. Pearson χ2 test was performed to compare proportions between two groups.

Q6)  The description of the follow up should be given before statistical analysis. Please, move lines between 137 and 142, before this paragraph.

--> We changed order of sentence as your recommendation.

Q7) Which was the minimum follow-up time to detect in-hospital deaths?

-->  We added the sentence like below;

The minimum and maximum follow-up time were 0 and 499 days after ECMO initiation, respectively.

Q8)   Line 144: Why did authors dichotomized all variables used in the Cox analysis? Cut-off values are used according to guide lines, if available, which represent standardized categories, i. e BMI classification of WHO. This will allow to compare results between studies.

--> We had used wrong terminology of dichotomized. We used categorized variables for Cox analysis.

Thus, we changed the sentence as below;

All variables tested in the Cox regression analysis were categorized. The SOFA score and initial diastolic blood pressure (BP) were dichotomized by their cut-off value for mortality. The cut-off was chosen based on the receiver operator characteristic curve and Youden index. History of diabetes defined when patients had a documented diagnosis for diabetes before admission or prescription for diabetes medication. According to the American Di-abetes Association, hyperglycemia is defined in hospitalized patients as a blood glucose level >140 mg/dL [23,24]. Diabetes is defined as a random plasma glucose level of ≥ 200 mg/dL with classic symptoms of hyperglycemia [25]. Therefore, we categorized various glucose parameters according to the definition of hyperglycemia (>140 mg/dL or >200 mg/dL) and evaluated the association between hyperglycemia and in-hospital mortality after ECMO support in patients with COVID-19. Body mass index (BMI) was classified as 3 categories (<18.5, 18.5–24.9, ≥25 kg/m2) according to the World Health Organization Asian classification [26].

Q9) Line 150-153: It is not a multivariate rather than a multivariable analysis. Please, modify.

--> We changed ‘multivariate’ into ‘multivariable’ as your recommendation.

Q10) Methods for goodness of fit of the model have to be reported and results in the Result section.

-->  Proportional hazard assumption was visually assessed using log(−log) plots. No variables included in the Cox regression analysis did not violate proportional hazard assumption. We also added survival curve of each Cox regression model in the supplementary Figure 1 and 2.

Results

Q11) Line 159-162: there is no indication of the distribution of the follow up time, it should be added.

--> We added the follow up time as below;

The median follow-up period was 40 (20-64) days.

Q12) Table 1 and 2: the distribution of patients’ characteristics is evaluated according to the outcome, a time-dependent variable, therefore comparisons between groups should be performed through appropriate statistical methods, such as Kaplan-Meier analysis. In this case, number at risk and number of cases at each time can be added. Moreover, if variables are used classified in categories, these should be reported in the descriptive analysis.

--> As your comment, we compared patient character and lab data according to survival which could be different by time. Even though survival could be affected by time, independent variables (patient character) were not affected by time. In case of lab data, vital sign or blood test results were measured at admission, thus those were also not time-dependent. Maximal glucose and minimal glucose could be affected by time. However, the main purpose that we compared difference between survivor and nonsurvivor group was to find covariate which should imputed in the multivariable Cox regression model. Thus, we moved table 1 and 2 which were not main findings of results into supplementary Tables.

Q13) Lines 266-267: Definition of preexisting diabetes is not clear, what authors mean by “diabetes medication use”? since when? How many prescriptions, etc.? Moreover, the definition should be moved to the Method section.

-->  As your recommendation, defining preexisting diabetes just by diabetes medication is not proper. Thus, we changed expression of preexisting diabetes into diabetes history. Diabetes history was defined as below;

History of diabetes defined when patients had a documented diagnosis for diabetes before admission or prescription for diabetes medication.

Q14)Table 3 will not be useful any more after an appropriate descriptive analysis.

-->  The main purpose that we compared difference between survivor and nonsurvivor group was to find covariate which should imputed in the multivariable Cox regression model. Thus, we moved table 1 and 2 which were rather redundant finding into supplementary Tables. We thought previous Table 3 are more relevant finding for making multivariable Cox regression models.

Q15) Line 130: substitute multivariate with multivariable or multiple and throughout the text.

-->  We changed multivariate into multivariable.

Q16) Lines 310-313: should be moved to the Statistical analysis section.

--> Those lines were move to the Statistical analysis section.

Q17) Lines 319-324: Move to the Statistical analysis section

--> Those lines were move to the Statistical analysis section.

Q18) Lines 332-334: Move to the Statistical analysis section

--> Those lines were move to the Statistical analysis section.

Q19) Table 4 and 5. Use only 1 cut off for small p-values, i. e <0.001. For those <0.05 give exact value.

--> We provided exact value when it was less than 0.05.

Q20) Supplementary Table 1, Table 4 and Table 5. All adjusted multivariable Cox models have estimates with low precision as indicated by the 95% confidence intervals. Authors should give indication of the power of the models. Were interactions in these models explored, such as between age, BMI and diabetes and/or glucose levels?

--> We added survival curves related with Supplementary table 1, table4, and table 5. Interactions between variables which inputted in the Cox regression models were evaluated with VIF. VIFs of glucose parameters with age and BMI were less than 1.10.

Discussion

Overall, this section needs to be revised and reorganized. Authors should avoid to repeat concepts already expressed in the other section of the manuscript. Results of the study can be compared and discuss related to similar studies. In particular:

Q21)  Lines 394-399: already used in the Introduction section, do not repeat.

--> We deleted those lines.

Q22) Lines 400-407: is redundant with results, it is not useful to repeat it.

--> We deleted those lines.

Q23) Lines 414-415: Results reported are not supported by those reported in the following lines 415-417.

-->We deleted those lines, since those were not support our results as you recommended.

Q24) Lines 422-423: the interpretation of the relative risk is not correct.

--> We deleted those lines, since those were not support our results as you recommended.

Q25) Lines 434-441: should be moved to methods section

-->  We moved those lines to methods section.

Q26) Lines 442-447: Remove. Results of the adjusted analysis can be discussed.

-->  We deleted those lines.

Q27) Lines 489-490: should be removed as redundant with Introduction section

--> We deleted those lines.

Q28) Lines 531-534: Authors should discuss how the mentioned limits could impact results for a better interpretation.

--> We added mention how the limits impact on results and future study plans like below;

This study has several limitations. First, this study had a small sample size; therefore, it was impossible to deduce a cause-effect relationship between hyperglycemia or hypo-glycemia and mortality. Thus, it is essential to validate whether hyperglycemia or hypo-glycemia are prognostic factor for in-hospital mortality of COVID-19 patients who under-went ECMO with another larger size cohorts as a future study. Second, there was signifi-cant missing data, since many patients were referred for ECMO under mechanical venti-lation. Missing data could make bias, thus our present results should validate with other cohorts. However, our results could show importance why clinicians should properly manage glucose parameters during ECMO as real-world data. Third, preexisting diabetes was defined only by a medical history of diabetes and diabetes medication without the results of HbA1c, since HbA1c was not available for all patients at admission. Fourth, the glucose parameters in our study could be fasting or random depending on the patient. These factors might act as biases; however, they reflect the real-world condition in which most blood samples in patients with COVID-19 were obtained regardless of fasting. For decreasing those biases, preexisting diabetes should be defined based on HbA1c and blood samples for measuring glucose parameters should be performed under fasting state in the future study.

Reviewer 2 Report (Previous Reviewer 1)

Authors have made significant changes on the draft and quality of presentation was improved.

It is very striking that significant changes on results have been made. For example, obesity is not an independent risk factor anymore, however hypoglicemia now has become a risk factor?.

There has been significant changes in authorship.

I believe that these major changes deserves a new submission.

Author Response

Q1) Authors have made significant changes on the draft and quality of presentation was improved. It is very striking that significant changes on results have been made. For example, obesity is not an independent risk factor anymore, however hypoglicemia now has become a risk factor?.

Answer; Previous manuscript contained data from 65 patients with COVID-19 treated with ECMO who enrolled in the registry of 21 hospitals in Korea between February 21 and December 31, 2020. Since the number of patients was small, the statistical power of previous manuscript was weak. Thus, reviewers had suggested that the number of patients should be increased for increasing statistical power. As reviewers’ comments, we reanalyze the data after enrolling more patients in our registry. The present manuscript contains data from 195 patients with COVID-19 treated with ECMO who admitted between February 2020 and May 2022 at 13 hospitals.

Present cohort was younger than previous cohort. The mortality and BMI of present cohort was higher than previous cohort. Even though present cohort was younger, and BMI was higher than previous cohort, the mortality was higher in present cohort. In previous cohort, BMI> 30 kg/m2 was a risk factor of mortality, but BMI>25 kg/m2 was not a risk factor. Most patients who was BMI >30 kg/m2 are younger than 65 years. Thus, we thought that obesity could be affected by age and obesity might be not a strong predictor of mortality in the previous cohort. The reason why severe obesity was not a risk factor in present cohort any more might be that previous cohort was older than present cohort.

Previous cohort

Present cohort

Number of patients

65

195

Number of participant hospital

21

13

Admission period

February 2020- December 2020

February 2020-May 2022

Mortality (%)

42.9%

54.6%

Mean age (yr)

62.37

59.74

Mean BMI

26.44

26.71

Even though obesity was not a risk factor of in-hospital mortality in the present cohort, hyperglycemia related variables were still a risk factor of in-hospital mortality in the present cohort. In case of hypoglycemia, the minimal glucose during hospitalization was not different between survivor and nonsurvivor groups in previous cohort, thus we did not perform further analysis. However, the minimal glucose level was significantly different between survivor and nonsurvivor groups in present cohort. Those discrepancy might be caused by difference of population between previous and present cohort. Pandemic peak could lead to changes in available medical resources which affect ECMO outcome as well as decision for indication of ECMO. In Korea, there was two hits of COVID-19 in September and December 2021. Those hits could affect ECMO outcome and patients’ characteristic difference between previous and present cohort. It is also a reason why mortality was increased in present cohort even though patients of present cohort were younger than previous cohort.

Q2) There has been significant changes in authorship.

Answer: The participating hospital are different between previous cohort and present cohort, thus the authorship was changed.

Reviewer 3 Report (New Reviewer)

Dear Authors,

Thank you very much for an opportunity to read and evaluate your study.

You conducted an interesting analysis and your data are comprehensive, however you presentation needs to improve significantly.

First, you paper must be concise, at current form it is way too long.

Introduction should be limited to one page and lead to your hypothesis. Additional information is redundant in this place.

Methods. Which part of data were collected prospectively and which retrospectively?

Results. Please, preset only main results and the rest should be included in appendix

Discussion. Please follow proposed pattern

1.Main findings

2. How do your results compare to other similar studies?

3. Clinical significance

4. Limitations

5. Conclusion

Please avoid rep[eating results in your discussion

Do not include in your discussion well known facts, which can be just referenced in context of your hypothesis and findings

Author Response

Q1) Introduction should be limited to one page and lead to your hypothesis. Additional information is redundant in this place.

--> We shortened the introduction part.

Q2) Methods. Which part of data were collected prospectively and which retrospectively?

--> Previous manuscript was little bit confusedly described about data collection. The present analysis was performed with data from Korean COVID-19 ECMO registry. The data which imputed in the registry was collected prospectively or retrospectively according to the time point. When patients were enrolled after discharge, data was collected retrospectively. When patients were enrolled before discharge, data was collected either retrospectively or prospectively. 

Q3) Results. Please, preset only main results and the rest should be included in appendix

--> As your recommendation, we moved Table 1 and 2 into supplementary section and decreased contents in the results section.

Q4) Discussion. Please follow proposed pattern

1.Main findings

  1. How do your results compare to other similar studies?
  2. Clinical significance
  3. Limitations
  4. Conclusion

--> We modified discussion as your recommendation.

Q5) Please avoid rep[eating results in your discussion

--> We deleted repeating of results in the discussion section.

Q6) Do not include in your discussion well known facts, which can be just referenced in context of your hypothesis and findings

--> We modified discussion as your recommendation.

Reviewer 4 Report (New Reviewer)

The authors in their paper report an elegant analysis on the role of hyperglycaemia in patients admitted for COVID-19. They observed that patients with severe hyperglycaemia or hypoglycemia before ECMO initiation is a negative prognostic facor. The information reported could be of interest for the readers of the journal, in particular for those involved in the care of patients with severe COVID-19.

I have the following concerns

Major points 

1) Hyperglycaemia was not associated with in-hospital outcome after adjustment for all the potential confounders. This point should prompt that hyperglycaemia is associated with outcome, but perhaps this is not a so strong prognosticator

2) Poor glycemic control is associated with RV subtle systolic dysfunction. At the same time, RV subtle dysfunction is a known consequence of COVID infection, with a possible relation with long COVID. This point should be discussed (doi:10.1161/CIRCIMAGING.120.012166).

3) Amongst the pathophysiology underlying the results, oxydative stress shoumentioned

Minor points

1) Diabetes is strongly associated with COVID related risk of death. However, diabetes is more common amongst elderly patients, usually with more severe comorbidities. Indeed, previous reports showed that only elderly with diabetes have a higher risk of death (https://doi.org/10.1210/clinem/dgab668) this point should be discussed.

2) The results would benefit from a validation cohort

3) Please report the cause of death amongst ECMO patients

4) Please report the share of missing for each variable 

Author Response

Q1) Hyperglycaemia was not associated with in-hospital outcome after adjustment for all the potential confounders. This point should prompt that hyperglycaemia is associated with outcome, but perhaps this is not a so strong prognosticator

Answer:  

We added several sentences in the discussion as your recommendation like below;

A complete explanation of why a glucose level >200 mg/dL before ECMO was the most significant risk factors of in-hospital mortality among other glucose parameters was not possible, as our study is a registry study. It appeared that glucose level >200 mg/dL before ECMO most sensitively reflected the stress hyperglycemia condition in patients with COVID-19 supported by ECMO, among other glucose parameters. It is also known that all types of artificial extracorporeal circulation devices, including ECMO, induce increased pro-inflammatory signaling pathways by initiating blood interactions with artificial sur-faces [49,50]. Moreover, an animal study showed that ECMO circulation leads to increased levels of pro-inflammatory cytokines, such as TNF-α and IL-6, especially in diabetic animals than in animals with normal glucose levels [51]. Thus, it is possible that patients with higher glucose levels before ECMO were more affected by ECMO, which induced a greater increase in inflammatory cytokines than patients with normal glucose levels be-fore ECMO. In other words, the glucose level >200 mg/dL before ECMO suggests that patients may be in a more vulnerable state in which  ECMO-induced inflammation impacts more severely than in patients who have blood glucose levels lower than 200 mg/dL. In another aspect, hyperglycemia was associated with in-hospital mortality, however it might not be a strong prognostic factor, since other hyperglycemia variables such as glucose levels >200 mg/dL before ventilator was not associated with in-hospital mortality after adjustment for age, BMI, and RRT before ECMO. It is known that right ventricular (RV) systolic and diastolic dysfunction is associated with poor glycemic control in type 2 diabetes [52]. RV dysfunction which evaluated with RV longitudinal strain (RVLS) was related with mortality in patients with COVID-19 [53]. Thus, it is also possible that hyper-glycemia could be an indirect risk factor for in-hospital mortality which related with RV dysfunction. Since we did not collect echo data in this study, it should be needed to evaluate whether hyperglycemia directly affect in-hospital mortality or indirectly affected through RV dysfunction in the future study.

Q2) Poor glycemic control is associated with RV subtle systolic dysfunction. At the same time, RV subtle dysfunction is a known consequence of COVID infection, with a possible relation with long COVID. This point should be discussed (doi:10.1161/CIRCIMAGING.120.012166).

Answer:  

We added several sentences in the discussion as your recommendation like below;

It is known that right ventricular (RV) systolic and diastolic dysfunction is associated with poor glycemic control in type 2 diabetes [52]. RV dysfunction which evaluated with RV longitudinal strain (RVLS) was related with mortality in patients with COVID-19 [53]. Thus, it is also possible that hyperglycemia could be an indirect risk factor for in-hospital mortality which related with RV dysfunction. Since we did not collect echo data in this study, it should be needed to evaluate whether hyperglycemia directly affect in-hospital mortality or indirectly affected through RV dysfunction in the future study..

Q3) Please report the cause of death amongst ECMO patients

Answer:

 We added cause of death in the results as your recommendation like below;

Cause of death was multiorgan failure (n=22), neurological complication (n=3), septic shock (n=58), and others such as bleeding or lung fibrosis (n=27).

Q4) Please report the share of missing for each variable 

As your recommendation, we reported missing data frequency in the supplementary Table 1.

 Data of patient who referred was incomplete. Little’s missing completely at random (MCAR) test was performed and most variable except variables in table 4 and 5 were likely missing at random.

We performed missing data handling with List-wise deletion. We added those in the method section like below;

There was incomplete data, especially in data from patients who referred. Number of missing data was reported in supplementary Table 1. Missing data was handled by List-wise deletion.

In addition, missing data could make bias, thus we added limitation of our study in the discussion as below;

Second, there was significant missing data, since many patients were referred for ECMO under mechanical ventilation. Missing data could make bias, thus our present results should validate with the other cohort. However, our results could show importance why clinicians properly manage glucose parameters during ECMO as real-world data.

Variable

Number of missing

Variable

Number of missing

Male sex

0

BT

7

Age

0

WBC

0

Hight

0

BUN

2

Weight

0

Creatinine

12

BMI

0

AST

4

Hypertension

0

ALT

4

Preexisting diabetes

0

Initial glucose

16

Chronic renal

disease

0

Glucose before

ventilator

53

Cerebral disease

0

Glucose before

ECMO

55

Cancer

4

Max glucose

53

Heart failure

0

Min glucose

56

Ischemic heart

disease

0

SOFA

53

Dyslipidaemia

0

PaO2/FiO2 ratio

before ECMO

16

Arrhythmia

0

Ventilator to ECMO >7 days

4

COPD

0

Cardiac arrest before ECMO start

8

Smoking

0

Inotropic use before ECMO

12

Initial systolic BP

6

Steroid use

14

Initial diastolic BP

6

RRT before ECMO

9

HR

6

Mortality

0

Round 2

Reviewer 2 Report (Previous Reviewer 1)

Authors have improved the quality of presentation and have justified all the significant changes that they have performed in the draft.

Author Response

Q1) Authors have improved the quality of presentation and have justified all the significant changes that they have performed in the draft.

--> We are appreciated your precise review.

Reviewer 4 Report (New Reviewer)

Thank you for your efforts in addressing my comments. However, I still have some suggestions:

Q1) I agree with the authors' reply.

Q2) The point raised by the authors is relevant. However, in these times we are facing predominantly with post-COVID patients. Therefore, I think that some discussion regarding post-COVID RV dysfunction and glucose levels disregulation should be reported (doi:10.1161/CIRCIMAGING.120.012166 and doi: 10.1016/j.pcd.2022.05.009). 

Q4) The lack of information regarding glucose levels in over 50 patients before ECMO should be acknowledged as a limitation

Author Response

Q1) I agree with the authors' reply.

--> We are appreciated your precise review.

Q2) The point raised by the authors is relevant. However, in these times we are facing predominantly with post-COVID patients. Therefore, I think that some discussion regarding post-COVID RV dysfunction and glucose levels disregulation should be reported (doi:10.1161/CIRCIMAGING.120.012166 and doi: 10.1016/j.pcd.2022.05.009). 

-->We added several sentences related with your comments and references like below;

Since we did not collect echo data in this study, it should be needed to evaluate whether hyperglycemia directly affect in-hospital mortality or indirectly affected through RV dysfunction in the future study. Moreover, recent meta-analysis showed that incidence of diabetes in post-COVID-19 patients was higher than healthy control subjects [54]. Among patients who recovered from COVID-19, 42% showed subclinical RV dysfunction which evaluated with RVLS [55]. Thus, it might be possible that hyperglycemia is associated with RV dysfunction even in the recovery period of COVID-19 or post COVID-19 syndrome. Those possible relationships should be evaluated in the future study.

Q4) The lack of information regarding glucose levels in over 50 patients before ECMO should be acknowledged as a limitation

-->We added your recommendation as below;

Second, there was significant missing data, since many patients were referred for ECMO under mechanical ventilation. Especially, there were significant missing data related with glucose parameters which is a main limitation of our study. Missing data could make bias, thus our present results should validate with other cohorts. However, our results could show importance why clinicians should properly manage glucose parameters during ECMO as real-world data.

This manuscript is a resubmission of an earlier submission. The following is a list of the peer review reports and author responses from that submission.

Round 1

Reviewer 1 Report

This multicentric observational study based on a national registry, included sixty-three patients with COVID-19 treated with ECMO in Korea between February 21 and December 31, 2020. The aim of the study is to indentify independent predictors of mortality among these patients.

The objectives, design and methodology of the study are correct, just like the results but it is not well-written. There is a lot of redundant information that you can find on Tables and it is also showed in the text, thus it should be summarized. For example:

  • Table 1 description (lines 180 to 187)
  • Table 2 description (lines 192-206)
  • Table 3 could be removed
  • Table 4 description (lines 243-256)
  • Table 5 description (lines 277-290).

In the discussion section, in lines 384-399, information was previously shown in the results section

Lines 421-425 are also redundant.

Therefore, a significant part of the text should be re-written in a shorter way, avoiding redundant information.

Reviewer 2 Report

This study aimed to evaluate whether preexisting diabetes, hyperglycemia before and during  ECMO support, and obesity were risk factors for mortality in patients with COVID-19 supported with ECMO in Korea. Although the study refers to the first year of the pandemic, it could bring important results for the management of severely ill patients with COVID-19.

Still, several aspects should be clarified and statistical analysis needs an important revision. As such, the manuscript should be rewritten and I suggest authors to include a biostatistician in the revision of the study and to consider an extension of the study period.

The following drawbacks should be addressed to improve the paper:

Material and methods

  • A short description of the Korean COVID-19 ECMO registry is needed giving indication of the population covered, the eligibility criteria. Are patients enrolled from all Korean hospitals?
  • Details on ECMO treatment are necessary: duration, discontinuation, related complications
  • Which was the last registry update?
  • The study outcome should be better explain: is it in-hospital mortality? At which time was it considered? If patients were discharged to other hospital, how were they considered?
  • The minimum follow-up time to detect in-hospital deaths should be reported
  • Description of the exposure/exposures is missing. Authors have to define exposure and give details how is it measured. The use of classifications should be justified based on the purpose of the study
  • The statistical analysis should be completely revised. The distribution of the variables should be checked in order to orient the proper statistical analysis, parametric or non-parametric one. The comparison between non-survivors and survivors should be made considering the outcome as a time-dependent variable. It is not clear why authors use several classifications of the glucose (tertiles of the distribution in addition to two different cut-off). The classification of the exposure factors should be made according to the aim of the study and the analysis should follow the objective. The association between the outcome and exposure with different classification can’t be evaluated by separate multiple models. If authors are interested in the severity of glucose levels, than glucose can be classified in a variable with more than two categories. In addition, the different factors analyzed, as initial glucose, glucose before ECMO, glucose during hospitalization, are probably correlated. Estimating the association between one of those with the outcome in separate models gives a distorted estimation of the parameter (i.e HR for Max glucose) as it doesn’t take into consideration the presence of the other factors (i.e. Initial glucose). Furthermore, evaluating 12 separate models increases the probability to detect a significant association only by chance. In fact, the significance level (α) is not anymore 5%. Therefore, after defining exposure factors use one multiple model to estimate their association with the outcome, adjusted for potential confounders. The procedure of the choice of the final model should be explained. Finally, model goodness of fit, collinearity and proportional hazards assumption should be checked and reported in the manuscript.

All above requires adequate sample size to guarantee adequate power of the model. Hence, considerations regarding the sample size should be given (number and method used to calculate it following the aim of the study and the methods considered to answer it). If possible, take into the consideration an extension of the study period allowing to increase the sample size.

  • Information on data completeness is also needed. If missing data occurred, a description on how they were handled and respective methods should be given.

Results

  • As the outcome is the mortality, give an estimate of the cumulative probability of death (and relative Confidence Interval); the only number and proportion of deaths in the sample could not be of interest.
  • Use proportion or percentage instead of ratio and SD in brackets instead of ± throughout the text.
  • Give indication of the overall follow-up duration and in each group.
  • If tables contain p-values of different statistic tests, report at the end of the table the test used.

Discussion

  • It should be rewritten according to the results and shortened with respect to the actual version that is too long.
  • Paragraph from line 384 to 399 is redundant with results; paragraphs from 319 to 330 and from 452 to 464 should be synthetized. It is should be useful to compare results of the study with those of other studies and discuss in relation to differences or similarities.
  • Paragraph from line 376 to 383 should be moved to methods.
  • Comparisons between predictors of the outcome evaluated using separate models cannot be made as not supported by the analysis.